# ZIKV Infection and miRNA Network in Pathogenesis and Immune Response

**DOI:** 10.3390/v13101992

**Published:** 2021-10-04

**Authors:** Carolina Manganeli Polonio, Jean Pierre Schatzmann Peron

**Affiliations:** 1Neuroimmune Interactions Laboratory, Department of Immunology, University of São Paulo, São Paulo 05508-000, Brazil; cmpolonio@usp.br; 2Laboratory of Neuroimmunology of Arboviruses, Scientific Platform Pasteur-USP (SPPU), University of São Paulo, São Paulo 05508-020, Brazil; 3Immunopathology and Allergy Post Graduate Program, School of Medicine, University of São Paulo, São Paulo 01246-000, Brazil

**Keywords:** microRNAs, ZIKV, flavivirus, viral infection

## Abstract

Over the years, viral infections have caused severe illness in humans. Zika Virus (ZIKV) is a flavivirus transmitted by mosquito vectors that leads to notable neurological impairment, whose most dramatic impact is the Congenital ZIKV Syndrome (CZS). ZIKV targets neuronal precursor cells leading to apoptosis and further impairment of neuronal development, causing microcephaly, lissencephaly, ventriculomegaly, and calcifications. Several regulators of biological processes are involved in CZS development, and in this context, microRNAs (miRNAs) seem to have a fundamental role. miRNAs are important regulators of protein translation, as they form the RISC silencing complex and interact with complementary mRNA target sequences to further post-transcriptional repression. In this context, little is known about their participation in the pathogenesis of viral infections. In this review, we discuss how miRNAs could relate to ZIKV and other flavivirus infections.

## 1. Introduction

Zika virus (ZIKV) is an arbovirus of the Flaviviridae family first isolated in 1947 from sentinel *Rhesus* sp. monkeys and *Aedes africanus* mosquitoes at the Ziika forest in Uganda, Africa [1]. ZIKV has a symmetrical structure, enveloped icosahedral nucleocapsid, and a 10 Kb (+) single-stranded RNA genome that encodes three structural proteins: Capsid (C), Pre-membrane (Pr-M), and Envelope (Env), and seven non-structural proteins: NS1, NS2a-2b, NS3, NS4a-4b and NS5 [2,3].

Previously, the potential for a viral outbreak was neglected, as it caused rare and mild infections in humans in Africa and Asia [4]. Later, outbreaks were identified elsewhere, such as Yap Island in Micronesia [5], French Polynesia in 2014 [6], Tahiti in 2013 [7], and New Caledonia in 2014 [8]. The main signs and symptoms known at the time included fever (37.8–39.5 °C), headaches, arthralgia of the hands and feet, conjunctivitis, and skin rash [5].

As of the first half of 2015, a generalized ZIKV epidemic occurred in South and Central America, with the northeast of Brazil being the most affected region. It is known that ZIKV causes Guillain-Barré Syndrome in adults [9] and congenital ZIKV syndrome (CZS) in fetuses infected during pregnancy, leading to severe neurological complications, including microcephaly, lissencephaly, ventriculomegaly, and cortical calcifications, associated or not with arthrogryposis, intrauterine growth restriction (IUGR), uveitis and retinal degeneration [10,11,12]. Unfortunately, most of these neurological modifications are irreversible and occurring in 6–12% of infected pregnant women [13,14]. Furthermore, neurological complications were also demonstrated in babies born without microcephaly [15,16,17].

Genetic differences, mainly related to antiviral immune response and neurodevelopment may directly influence susceptibility to infection [18,19]. Many of these genes code for proteins involved in signaling pathways, as adaptor proteins and transcription factors. Importantly, these molecules may be a target of post-transcriptional or post-translational regulation when microRNAs (miRNAs) may have a fundamental role. The regulatory capacity of miRNAs has been described over the years in several animals and plants species [20]. Their participation has been studied in different diseases, such as cancer [21,22,23], diabetes [24,25], multiple sclerosis [26,27,28], and viral infections, including infections by neurotropic flaviviruses [29,30]. 

Although the Brazilian Ministry of Health declared the end of the Public Health Emergency of National Importance (ESPIN) caused by ZIKV in 2017, neurological impairment leads to serious consequences for children, making them dependent and hindering their development and insertion into society. Therefore, understanding molecular mechanisms of susceptibility provides important knowledge for the development of vaccines or therapeutic interventions. In this review, we discuss how miRNAs may regulate important mRNAs to influence the outcome of ZIKV infection.

## 2. ZIKV and Neurological Impairment

ZIKV has been proven to cause neurological impairment by several studies using human samples and different experimental models. The first major evidence that ZIKV causes microcephaly came in 2015 [31]. The *post-mortem* analysis of an infected fetus indicated IUGR, numerous cortical and subcortical calcifications, moderate ventriculomegaly, and 26 cm brain perimeter, indicating microcephaly (control ≤ 32 cm ± SD), associated with the presence of ZIKV RNA in the brain, suggesting a neurotropism. In addition, the histological and cellular analysis showed astrogliosis in the subarachnoid space related to viral particles present in neurons. Following studies corroborated these findings and demonstrated the presence of ZIKV in the amniotic fluid [32], placenta [33], cerebrospinal fluid (CSF) [34], and retinas [35] of babies with microcephaly. 

Three studies were pioneers demonstrating susceptibility to infection and neurological damage [36,37,38]. One study used pregnant females, which comprised of interferon alpha/beta receptor subunit 1 (IFNAR1)^−/−^ or wild type (WT) mice treated with IFNAR1 receptor blocking antibodies (clone MAR1-5A3). The mice were infected subcutaneously with 10^3^ focus forming units (FFU) at embryonic day (E) E6.5 and E7.5 and pups were analyzed at E13.5 and E15.5 showing fetal resorption, IUGR, pallor, presence of necrotic tissue, and ZIKV in pups’ placenta and brain tissue analyses [36]. Another group performed ZIKV infection via cerebroventricular route at E13.5 and analyzes at E16.5, which showed that ZIKV infected ventricular and subventricular zone where most neuronal precursors cells (NPCs) are found [37]. A third study was published by our group [38], demonstrating that WT pups born from pregnant Swiss James Lambert (SJL) infected mice presented IUGR characterized by reduced weight, size, length, height, and biparietal measurement. Histological analysis showed a decrease in the cerebral cortex, with nuclear vacuolization, chromatin marginalization in neurons from the cortex, thalamus, and hypothalamus, which was not observed in the cerebellum and hippocampus. 

Monkey models have further ratified the neurological damage caused by ZIKV. Rhesus and adult cynomolgus monkeys were susceptible to infection, as viral RNA was found in plasma, saliva, CSF, brain, female and male reproductive tracts, semen, and transiently in vaginal secretions [39]. It has also been shown that non-pregnant and pregnant Rhesus monkeys remain viremic for 21 and up to 57 days after infection [40]. Also, the virus was persistent in the CNS and lymph nodes due to positive regulation of mammalian target of rapamycin (mTOR), pro-inflammatory and anti-apoptotic signaling pathways, as well as negative regulation of extracellular matrix signaling pathways [41].

Vertical transmission-related studies showed that intravenous or intra-amniotic infection during the second trimester-pregnant monkeys leads to high levels of ZIKV in placental and fetal tissues, especially in the brain, which exhibited calcifications and reduced numbers of NPCs [42]. Infected monkeys at the beginning of pregnancy exhibited more complex neuropathies, such as cerebral microcalcifications, hemorrhage, necrosis, vasculitis, gliosis, and NPCs apoptosis, and also magnetic resonance analyses indicated damage to the deep gray substance [43]. In addition, abnormal oxygen transport within the placenta has been observed as a result of uterine vasculitis and placental villous damage caused by ZIKV [44].

These results clearly showed that ZIKV crosses the placenta and has a tropism for the fetal brain, leading to important tissue damage. It has been shown that ZIKV infects NPCs and neurons, both in vitro and in vivo, inducing cell death by autophagy and apoptosis, as evidenced by active caspase-3 expression analysis. In addition, the expression profile of 88 cell death-related genes showed that ZIKV positively regulates genes such as *Bmf*, *Irgm1*, *Bcl2*, *Htt*, *Casp6,* and *Abl1*, and negatively *Gadd45a*, *Tnfrsf11b*, *Fasl*, *Atg12*, *Bcl2l11,* and *Dffa* in the brain of SJL neonates born from ZIKV infected pregnant mice compared to uninfected mothers [38]. Moreover, a reduction in the cortical layer was observed due to the death of NPCs TUJ^+^, SOX-2^+,^ and TBR-1^+^, specific markers of NPCs that originate neurons and glial cells, evidenced by in vitro studies using cerebral organoids or mini-brains [38,45,46].

Although NPCs and neurons are the most affected cells by ZIKV infection, astrocytes deserve attention in their participation during its pathogenesis [47]. Even though little is known about the role of infected glial cells in CZS development, it has been shown that astrocytes can be infected by the virus and may play a fundamental role in the pathogenesis of microcephaly [48].

Reactive astrocytes have been identified in fetuses at 32 weeks of gestation in the brain regions affected by the virus [31]. The impaired functioning of astrocytes contributes significantly during microcephaly development, as it affects cortex growth due to impairment in neurogenesis and gangliogenesis [49,50]. Astrocytes infected by ZIKV present dysregulation in protein translation, glucose metabolism, synaptic control as well as in cell migration and differentiation [47,51,52]. Moreover, it has been shown that astrocytes are more susceptible to infection by ZIKV than neurons since they tolerate greater viral loads, suffer less apoptosis, and, consequently, allow a greater viral replication [53].

## 3. MicroRNAs

miRNAs are small non-coding and regulatory single-stranded RNA molecules that perform post-transcriptional regulation of mRNAs sequences by binding to 3′or 5′ untranslated regions (UTR) that destabilize and block translation of encoding proteins [54]. It is known that more than 60% of human genes that encode proteins have at least one conserved miRNA binding site, and several non-conserved sites [55,56]. Therefore, miRNAs expression must be well controlled, since their dysregulation is associated with pathologies, including those associated with neurological development [57].

miRNAs are transcribed and processed (Figure 1) from long primary chains (pri-miRNA) with more than a thousand nucleotides [54]. They are processed in the nucleus by a type III endoribonuclease (Drosha) and its cofactor DiGeorge Syndrome Critical Region Gene (DGRC8) generating the shorter precursor miRNA (pre-miRNA) with approximately 65 nucleotides long [58,59]. Following, the pre-miRNA is transferred to the cytosol by exportin 5 protein [60,61,62]. In the cytosol, pre-miRNA is processed by RNase III (Dicer) and RNA binding cofactor TAR RNA-Binding Protein (TRBP) in a short double-stranded miRNA with 20–24 base pairs, known as miRNA duplex [63]. The Argonaute (AGO2) is further recruited [64] forming the miRNA-containing RNA-induced silencing complex (miRISC) [65], where the mature miRNA (guide strand) [63,66] pairs with several complementary mRNA sequences to interfering in encoded proteins production [67].

The mechanisms of miRNAs action are divided into (1) target transcript destabilization, (2) translation inhibition, and (3) transcriptional silencing. Destabilization is represented by a) deadenylation followed by mRNA 5′CAP removal, once AGO2 is associated with GW182, which recruits CCR4-NOT complex and promotes the removal of poly-A tail, leading to mRNA destabilization and further mRNA degradation by exoribonucleases [68]. The translation inhibition is represented by (a) AGO competition for 5′CAP, once miRNAs lead AGO2 to mRNA, which competes for the 5′CAP, preventing mRNA and ribosomes association [69,70]; (b) blocking translation initiation, preventing poly-A tail interaction with poly-A binding protein C1 (PABPC1) and 5′CAP interaction with Eukaryotic Translation Initiation Factor 4E/4G (eID4E/eIF4G) [71]; and (c) dissociation of ribosomes, in which some miRNAs lead to early disassembly of ribosomes [72]. Finally, miRNAs perform transcriptional silencing that involves RNA processing bodies (*p*-bodies) that are cytoplasmic ribonucleoprotein aggregates [73]. miRNAs directly target mRNAs to *p*-bodies, where they are temporarily and reversibly repressed or destabilized [74].

This provides evidence for the complex translational regulation performed by miRNAs, which are important in different scenarios as regulatory molecules for several biological processes, making them biomarkers for detection and progression of diseases, as well as a target for therapeutic intervention [75].

## 4. Neurodevelopment-Related miRNAs and ZIKV Infection

ZIKV leads to irreversible neurological damage and, in this context, the study of molecules that interfere in the neurodevelopment becomes of great importance during CZS development. miRNAs have also been studied during embryonic development. Interestingly, Dicer deficient animals are not viable and suffer spontaneous abortion around the seventh day of gestation. Currently, conditional mice are available, in which Dicer deletion is performed by Cre-lox recombination. Using this system, it was possible to demonstrate that miRNAs are involved in neurodevelopment since dopaminergic neurons without Dicer undergo progressive apoptosis [76]. In addition, Dicer deficient mice are not able to generate viable embryonic stem cells (ESC) and maintain this population during mouse development [76].

Conversely, studies are demonstrating the importance of miR-9 for brain development [77], radial glia proliferation, and also neuronal and glial differentiation [78]. MiR-9 is highly conserved in all mammals, and, in rodents, it is specifically expressed in the brain, mostly in NPCs during neuronal differentiation. MiR-9 KO mice have smaller brain hemispheres and olfactory bulbs compared to WT animals, associated with decreased cerebral cortex and ventriculomegaly, characteristics similar to microcephaly [79,80]. In addition, it has been shown that cortex neuronal differentiation involves intermediate progenitor cells specificity and their development is regulated by several miRNAs, including miR-9 [81,82].

There are several genes related to autosomal recessive microcephaly (MCPH-Microcephaly Primary Hereditary). There are 12 MCPH loci (MCPH1-MCPH12) that have been mapped and contain the following genes: *Microcephalin*, *WDR62*; *CDK5RAP2*; *CASC5*; *ASPM*; *CENPJ*; *STIL*; *CEP135*; *CEP152*; *ZNF335*; *PHC1,* and *CDK6*. It is believed that these genes lead to disease phenotype due to premature chromosomal condensation, damaged DNA, disturbed microtubule dynamics, transcriptional control, and hidden centro-somatic mechanisms that regulate the number of neurons produced by neuronal precursor cells [83,84].

Interestingly, host RNA-binding proteins interact with viral genome UTRs regulating viral replication and translation [85]. An important regulator of NPC development, the RNA-binding protein Musashi-1 (MSI1), interacts with the ZIKV genome and unexpectedly improves viral replication [86]. It was also shown that during neurodevelopment, Musashi-1 interacts with MCPH-1 3′ UTR to regulate its expression to normal brain functioning. Interestingly, during ZIKV infection, Musashi-1 binds to the ZIKV 3′UTR genome to enable viral replication, and consequently, leads to impaired neurodevelopment. Recently, through computational analysis, it has been hypothesized that the ZIKV genome activates six host miRNAs that result in modifications based on neuronal genetic pathways that share significant mutual homologies with the 12 MCHP genes [83].

The importance of miRNAs during ZIKV infection has been described by several groups. A Brazilian group, using human neurospheres, showed that ZIKV replication stops the proliferation and differentiation of neuronal cells [87]. In addition, ZIKV upregulates a target network related to viral replication and downregulates molecules associated with the cell cycle and neuronal differentiation. Still in this work, it was demonstrated that ZIKV infection modulates pathways involved in RNA processing (DDX6, PCBP2), miRNAs biogenesis (DGCR8, XPO1), regulation of translation initiation (eIF3c), and proteins, such as splicing factors (SFPQ, PRP8), ribosomal proteins (RPS6KA5, RPL28), proteins related to innate immune response (TLR4) and neuronal development (NEUROD1, SATB2). 

Interestingly, ZIKV capsid directly interacts with Dicer to improve infection. Using a capsid mutant (H41R) it was demonstrated that ZIKV loses the ability to inhibit neurogenesis and corticogenesis. Therefore, the interaction between ZIKV H41R capsid and Dicer is necessary for the pathogenesis and the characteristic lesions [88].

Moreover, the ZIKV envelope (E) modulates the expression of the host’s miRNAs and, using bioinformatics tools, it was shown that the pathways involved were related to cell cycle and development processes [89]. ZIKV E protein causes NSCs quiescence by increasing the number of cells in the G0/G1 cell cycle phase. Also, the E protein induces NSCs apoptosis, further impairing neuronal differentiation and migration. 

Recently, post-mortem analysis of brain tissue from babies born with severe CZS showed an upregulation of miR-145 and miR-148a, and in silico analysis indicated that their target genes are involved in neurodevelopment pathways, such as glial differentiation, neurogenesis, and cerebral cortex development [90].

In response to ZIKV infection, NPCs release extracellular vesicles containing miRNAs, such as miR-4792, that modulate genes related to oxidative stress and neurodevelopmental processes [91]. Through RNA-sequencing, miR-sequencing, and AGO-iCLIP-sequencing, it was shown that ZIKV increases miR-124 expression acting over TFRC mRNA to further cell proliferation attenuation [92]. In primary murine neurons, the Puerto Rico ZIKV strain (PRVABC59) modulates miR-29a, miR-124, miR-155, and miR-203 which have a role in increasing antiviral immune response and brain damage [93]. In human astrocytes, ZIKV infection increased miR-17-5p, miR-30e-3p, and miR-30e-5p expression, regulating genes involved in cell cycle and immune response [94] (Table 1). 

In addition, ZIKV genome sequencing also expresses 47 miRNAs that target pathways involved in cell signaling regulation, neurological functions, and fetal development assisting in the establishment of microcephaly caused by ZIKV [95]. However, ZIKV negatively regulates essential miRNAs during CNS development, neuronal and glial differentiation, and positively regulates miRNAs that degrade essential genes in the same processes for the normal functioning of the organism. This confirms the harmful role of ZIKV in the CNS. These findings highlight the importance of miRNAs during ZIKV infection, correlating it with the pathogenesis of microcephaly caused by the virus.

## 5. Antiviral Immune Response and miRNA

There are many studies emphasizing miRNAs importance during flavivirus infection. In some cases, miRNAs are found upregulated by infection leading to immune response inhibition [96] (Table 2). For example, the Japanese encephalitis virus (JEV), which infects neurons and microglia cells causing neuronal damage and inflammation, modulates miRNAs expression that downregulates genes involved in cellular immune response and antiviral genes [96]. The JEV infection increases miR-146a expression, which downregulates TRAF6, IRAK1, IRAK2, and STAT1 genes. In addition, miR-146a suppresses activation of NF-κB and Jak-STAT pathway, resulting in negative regulation of ISGs (IFIT-1 and IFIT-2) and facilitating viral replication [96]. Interestingly, it has been shown that NS3 from JEV degrades miR-466d increasing viral replication [97]. Moreover, miRNAs assist viral replication directly, such as miR-21, which promotes DENV replication in human liver carcinoma cells [98].

On the other hand, miRNA overexpression may be beneficial to the host during flaviviruses infections, leading to decreased viral replication. For example, miR-155 upregulation in CHME3 cells, a microglial cell line, results in a significant reduction of JEV replication by decreasing NF-κB pathway genes and STAT-1, at the same time increases microglial activation [99]. Other miRNAs favor antiviral immune responses, preventing virus spreading, such as during JEV infection that upregulates miR-19b triggering Ring Finger Protein 125 (RNF125) inhibition, a negative regulator of RIG-I signaling. Thus, miR-19b increases inflammatory response, including type I IFNs production, and decreases glial activation and neuronal damage [100] (Table 3).

Interestingly, miRNAs are very promiscuous. The same miRNA modulates different mRNAs from the same signaling pathway, while different miRNAs interfere in the same mRNA. In this way, miRNAs assume ambiguous roles under different situations. Therefore, some flaviviruses could use miRNAs to suppress the immune response and, consequently favor its replication, while other flaviviruses lead to host miRNAs expression that will favor the immune response against the virus.

In this context, it is known that ZIKV decreases the immune response, but so far, none of the described mechanisms involved miRNAs [101,102,103]. Hence, it is plausible to think that ZIKV modulates miRNAs that will modulate genes of the antiviral immune response. This may either impact viral biology or the pathogenesis of the infection.

## 6. Concluding Remarks

In summary, we highlight miRNAs as potential regulators of biological processes in face of viral infections, including ZIKV (Figure 2). Thus, they modulate essential pathways for brain development, neurogenesis, apoptosis, autophagy, as well as inflammatory and antiviral responses. Here, we showed many in silico studies describing modulated miRNAs and their targets. However, there is still a lack of knowledge about specific miRNA targets and their effects during ZIKV infection. Therefore, more in-depth research needs to be carried out to further add to the knowledge of ZIKV infection and microcephaly caused by vertical transmission, assisting in the development of future therapeutic interventions.

## Figures and Tables

**Figure 1 viruses-13-01992-f001:**
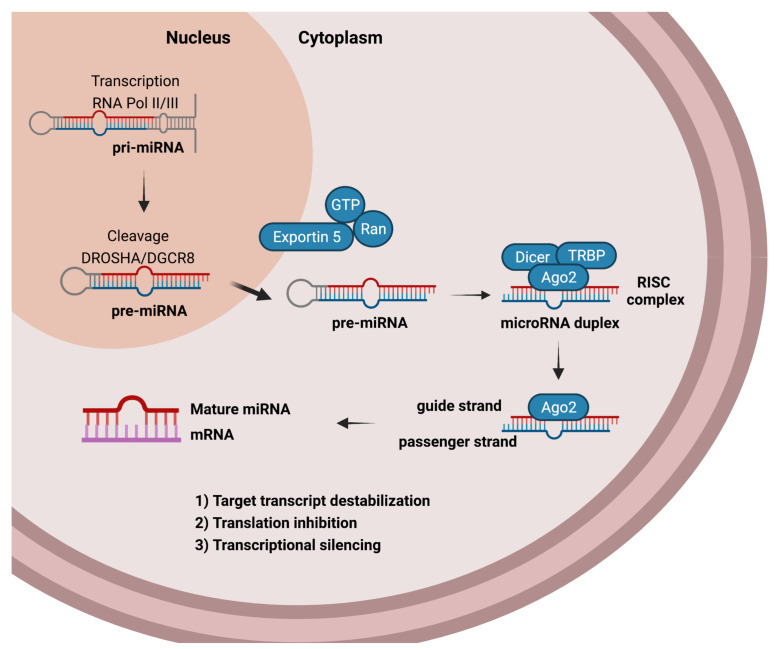
Overview of miRNA biogenesis. Biogenesis consists of the miRNA gene transcription and maturation that involves the following steps: Pri-miRNA processing by DROSHA; Processing of pre-miRNA by DICER; Other modifications to produce mature miRNA and RISC complex formation to further mature miRNAs regulate mRNA and consequently interfere in protein production. Figure created by authors using BioRender.com.

**Figure 2 viruses-13-01992-f002:**
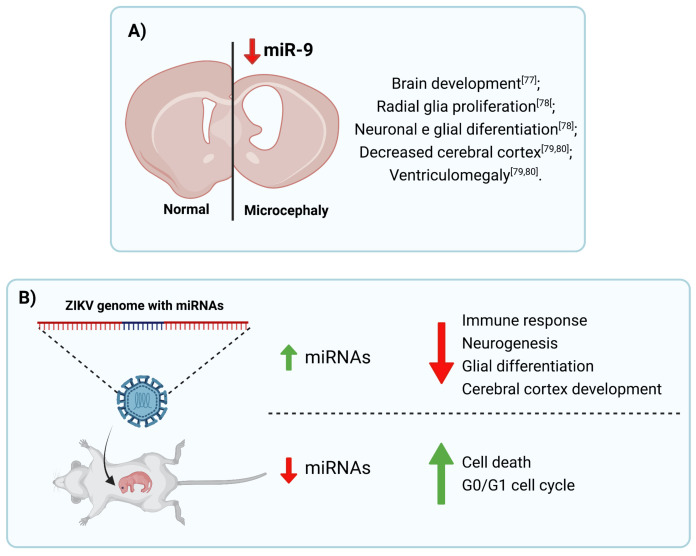
MiRNAs post-transcription regulation. miRNAs control (**A**) brain development and different pathway involved (**B**) in antiviral immune response, neurogenesis, cell death, and cell cycle in response to ZIKV and other flavivirus infections. Figure created by authors using BioRender.com.

**Table 1 viruses-13-01992-t001:** miRNAs and neurodevelopment.

miRNA	Virus	Cell Source
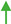 miR-9 [77,78,79,80]	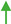 Neurodevelopment	Brain, mostly human NPC
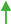 miR-145; miR-148 [90]	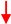 Neurodevelopment	Human brain tissue
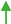 miR-4792 [91]	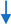 Neurodevelopment	Human NPCs
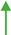 miR-29; miR-124; miR-155; miR-203 [92,93]	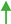 Brain damage	Primary murine neurons

**Table 2 viruses-13-01992-t002:** miRNAs that favors viral replication and decrease immune response.

miRNA	Virus	Immune Response Component
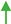 miR-146 [96]	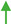 JEV replication	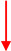 TRAF6; IRAK1/2; STAT-1NF-κB resulting a decreasing of IFIT1/2
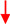 NS3 of JEVmiR-466d [97]	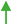 JEV replication	
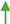 miR-21 [98]	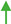 DENV replication	

**Table 3 viruses-13-01992-t003:** miRNAs that increase immune response.

miRNA	Virus	Immune Response Component
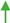 miR-155 [99]	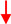 JEV replication	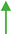 NF-κB and STAT-1 pathway genesMicroglial activation
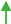 miR-19b [100]	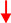 JEV replication	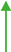 RNF124 and consequently increase RIG-I signaling.Type I IFN production

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
