# Peer review of "ZIKV Infection and miRNA Network in Pathogenesis and Immune Response"

_viruses, 2021, doi:10.3390/v13101992_

Round 1

Reviewer 1 Report

The article by Polonio and Peron is an interesting insight into the still poorly understood relationship between miRNAs and ZIKV infection,
with a very good focus on the neurological complications that made ZIKV sadly notorious.
Some weak points:
The chapter about miRNAs is absolutely too long and contains a uselessly detailed description of miRNA biogenesis. These information are well known to all people
working in molecular biology or virology field, and on the other hand are of little interest to people interested mainly in the clinical aspects of the topic.
I suggest to shorten the text and rely mainly on the figure (fig1) and the references for the details. Furthermore, the paragraph about miRNA function, though more synthetic,
contains also some mechanisms which are not present in humans, so they should probably be omitted.
The chapter about viruses, miRNAs and immunity creates some confusion in the readers: many works are cited, with no understandable logical order, mixing together
studies involving flaviviruses but also completely unrelated viruses. So, the first line (267) needs to be changed or the citations about other viruses should be omitted or moved 
elsewhere (in my opinion, better omitted; there are enough articles on miRNAs and immunity and on flavivirus and immunity, and also a few on miRNAs and flaviviruses, to make up
the chapter without nee of other unrelated viruses).
In general, this chapter is very important in the scope of this review and I think it should be extensively revised, with more logical order, maybe a summary table, and a more accurate choice of references
to provide a more structured track for supporting a richer paragraph of conclusions.
The reference section contains a huge number of mistakes showing it was probably prepared with insufficient care. In a review, in particular, this is not acceptable since
the reference section is of major importance and must be of adequate quality.

Some minor points:
- line 14: 'ZIKV targets'
- line 14-15: 'leading' is used twice in the same period
- line 26: 'first isolated' ; 'sentinel ....monkeys'
- line 29: ' Capsid (C)'
- line 31: ref. 2 doesn't seem particularly related to this part of the text, please find a more appropriate one
- line 36: 39,5°C is not a moderate fever
- line 45: 'and they occur'
- line 46: 'furthermore'
- line 59: 'serious consequences for children'
- line 60-63: Please rewrite the whole period in a more grammatically correct and meaningful way
- line 64: 'how miRNAs'
- line 69: 'of an infected fetus'
- line 76: liquor is not referred to in ref. 31. There is probably a mistake in reference number
- line 83: 'analyses'
- line 84: please define the abbreviation NPCs
- line 99: 'second trimester-pregnant monkeys'
- line 104: 'analyses'
- line 108: '... brain, leading to...'
- line 109: 'in vivo', in italic
- line 115: please define the abbreviations or explain briefly their biological context
- line 116: omit 'also corroborated by othe groups'
- line 126: please choose one between 'and' and as well as'
- line 128: '..since they tolerate...'
- line 123: ref 48 is a duplicate of ref 31. Please change numeration accordingly
- line 129: 'allow a greater viral replication'
- line 134: 'to destabilyze and block'
- line 137: 'since their dysregulation'
- line 144: 'by a type III endoribonuclease (Drosha), which presents RNaseIII domains...'
- line 149: 'shorter', 'approximately 65 nucleotides long'
- line 152: 'RNase III (Dicer)'
- line 162: fig. 1 contains two typos: 'nucleous' and 'exportina'
- line 169: 'in which'
- line 189: 'becomes'
- line 194: 'cre-lox'
- line 246: 'mir-148a' , 'their target genes'
- line 257: 'which regulates'
- line 268: 'miRNAs are found upregulated...'
- line 271; 'downregulates'
- line 272: the beginning of the sentence is a repetition of the previous one.
- line 282: which family of miRNAs?
- line 303: flaviviruses have nothing to do with the last cited articles, so this sentence if not appropriate.
- line 308: the last period of the chapter is written in a poorly understandable way, and also its scientific meaning is not very strong . 
Please rewrite in a more correct way, both from the formal and the biological points of view.
- line 314: 'including ZIKV'
- line 318: 'their effects'
- line 328: these are 'fundings', not 'acknowledgments'

- line 333: The article title is written in italic, and the journal name is missing
- line 337: the qualification note 'Ph.D' has been mistakenly added as if it was an author's name (twice). Furthermore, the 
journal name is missing
- line 339: The indication 'Pacific, N.' is mistakenly added as first author of the article
- line 343: the article title is written in upper case letters and the journal name and issue is written twice.
- line 348: there is a typo instead of '--'
- line 350-351: the qualification note 'Ph.D' has been mistakenly added as if it was an author's name. The journal name is missing
- line 353: The bibliographic entry is not understandable, please rewrite it correctly
- line 375: Please rewrite authors' names in accordance with the jounal's guidelines (style and number of names). Furthermore, there is
a typo (humans and 2 congenital...), and the journal name and issue are missing (the paper has recently been published)
- line 383: the journal name is missing
- line 385: the journal name is missing
- line 395: the journal name is missing
- line 411: Please take care to the style in authors' names
- line 414: The first author's name contains a '1'; the last cited name is written incorrectly; the article is written incorrectly;
the issue information is incomplete and the doi is incorrect.
- line 417: the authors names and the journal issue information are incorrect
- line 446: the first line of authors names is missing, as is the journal's name
- line 448: the atricle title is written twice
- line 482: authors' names are missing
- line 522: the journal's name is missing
- line 526: the term 'report' is not part of the paper's title
- line 529: part of the title and the journal's name are missing
- line 531: the term 'article' is not part of the paper's title
- line 542: please format the authors' names correctly
- line 546: the article's name in written incorrectly
- line 574: author's names, part of the title, the journal's name and doi are missing
- line 576: journal's name and issue anre missing
- line 583: some author's names, good part of the title and journal's name are missing

Reviewer 2 Report

The manuscript entitled “ZIKV infection and miRNA network in pathogenesis and immune response” attempts to provide support for implications of the miRNA effect on the impact of immune response and potential adverse pathogenesis. However, the manuscript is relatively poorly written and needs considerable editing in English.

Major comment: it appears that the manuscript is mostly a review of other studies, with little to no reference to the work that was conducted by the authors. For example, lines 67-76 refer to studies by the authors, but there is no:

  1. human use protocol for the study of fetal tissues;
  2. no indication of the number of fetuses or patients involved;
  3. no data for the fetal or patient tissues examined;
  4. no data for the outcome of the fetuses/patients.

There should be Tables for the Results.

Other comments:

  1. Line 36: use periods instead of commas for temperature.
  2. Line 52: spell out “miRNA” in its first usage in the body of the manuscript.
  3. Line 78: spell out IFNAR1-/- and WT during first usage. In addition to other acronyms throughout the manuscript, spell out when necessary.
  4. Figure 1: nucleous should be spelled nucleus. Pre-miR should be pre-miRNA.
  5. Line 246: I believe that “e” is Portuguese for “and”. Change to English.
  6. Delete or change Figure 2 to include how miR-9 impacts on micorcephaly and what miRNAs ae involved with cell death.

References. Many errors, e.g., 1) journals not listed, 2) Some citations is caps and others correctly in lower case. Review the journal citations and ensure that they are entered correctly.

Round 2

Reviewer 1 Report

The paper by Manganeli and Schatzmann has been revised following my indications in an appreciable way.

My major concerns have been addressed at least in part, although:

-The chapter about miRNA biogenesis has been shortened, but is still probably too long, and some more editing could have made it better.
-The paragraph about miRNA function has been modified very slightly, and mechanisms which are absent or of minor significance in humans are still present. The part which has been eliminated, curiously, was probably more appropriate than other mechanisms still presented in the text.
-The chapter about miRNAs and viruses, though extensively revised, retains the presence of viruses which have nothing to do with flaviviruses
 (VSV, PRRSV), and in my opinion could be still somehow improved.

I think in these three sections there can be a further improvement with not excessive workload, so I will propose the editors a further minor revision.

Some minor points that need to be fixed:

- line 49: 'many of these genes code'
- line 79: females was correct, not females'
- line 85: 'analyses', not 'analyzes'
- line 150: 'type III'. The same mystake was on the first submission. Drosha is a type III nuclease.
- line 156: omit 'with'
- fig.1 is shown twice (the older version and the newer, corrected one)
- Table 2: in contrast with the paragraph premises, VSV and PRRSV are not flaviviruses.
- line 336-337: please check for the correct use of commas
- line 433: there are still mistakes in this entry
- line 466: the first three authors' names are still missing
- line 494: the authors' names are still uncorrect
- line 532: the term 'report' is not part of the paper's title
- line 538: the term 'article' is not part of the paper's title

Reviewer 2 Report

Overall, the manuscript reads well.  A few grammatical errors that need to be addressed.

  1. Page 2, line 85. Change "analyzes" to "analyzed".
  2. Page 3, line 113. I believe that the word autophagy is not correctly separated. It should be "auto-phagy".
  3. Page 6, line 190. Change this to "This provides evidence for the complexity of translational ...".
  4. Page 9, line 305. Change to "... virus infections, leading ...".
  5. Page 10, lines 324. Change to "..., while other flaviviruses lead to host ...".
  6. Page 10, line 324. Change "potent" to "potential".
  7. References. Many errors, Authors need to pay special attention to detail.
    1. Reference titles in caps should be in lower case. #4, #8, #10, #13, #16 (also open should be Open", #17, #20, #23, #24, #26 (also period after "One."), #29, #31, #36, #37, #41, #43, #46, #53, #66, #71, #73, #80, #82, #83, #84, #90, #92, #94, #95 (MicroRNA should be in lower case), #99, #101, #104, #108, and #109. Also, virus should be in lower caps following Zika, to read "Zika virus".
    2. Reference 9, Guillain-Barre should be in caps. Also, the journal needs reformatting. Should be Euro. Surveill. (period after Euro and Surveill should be capitalized.
    3. Reference titles following a : should be in lower case. #12, #49, #54, #86, #105
    4.  Reference #33. Delete space "eff ects" to "effects".
